

# An optimized LSTM network for improving arbitrage spread forecasting using ant colony cross-searching in the K-fold hyperparameter space

Zeliang Zeng[1], Panke Qin[1,2], Yue Zhang[1], Yongli Tang[1], Shenjie Cheng[1], Sensen Tu[1], Yongjie Ding[1], Zhenlun Gao[1] and Yaxing Liu[1]

[1] School of Software, Henan Polytechnic University, Jiaozuo, Henan, China
[2] Hebi National Optoelectronic Technology Co, Ltd, Hebi, Henan, China

## ABSTRACT

Arbitrage spread prediction can provide valuable insights into the identification of arbitrage signals and assessing associated risks in algorithmic trading. However, achieving precise forecasts by increasing model complexity remains a challenging task. Moreover, uncertainty in the development and maintenance of model often results in extremely unstable returns. To address these challenges, we propose a K-fold cross-search algorithm-optimized LSTM (KCS-LSTM) network for arbitrage spread prediction. The KCS heuristic algorithm incorporates an iterative updating mechanism of the search space with intervals as the basic unit into the traditional ant colony optimization. It optimized the hyperparameters of the LSTM model with a modified fitness function to automatically adapt to various data sets, thereby simplified and enhanced the efficiency of model development. The KCS-LSTM network was validated using real spread data of rebar and hot-rolled coil from the past three years. The results demonstrate that the proposed model outperforms several common models on sMAPE by improving up to 12.6% to 72.4%. The KCS-LSTM network is shown to be competitive in predicting arbitrage spreads compared to complex neural network models.

## INTRODUCTION

With the rapid development of quantitative finance, a significant rise in algorithmic trading (AT) and high-frequency trading (HFT) has emerged as one of the most notable changes in the current financial markets (*Malceniece, Malcenieks & Putniņš, 2019*). Statistical arbitrage, as a prominent form of algorithmic trading, has gained widespread recognition and application in both academia and industry. Compared with other investment tools, arbitrage trading offers stable yields and lower risk. However, the proliferation of competitive arbitrageurs results in higher execution risk (*Kozhan & Tham, 2012*). The combination of arbitrage strategies and forecasting methods can facilitate the identification of additional trading opportunities and reduce the execution risk. Unfortunately, the

Corresponding author
Panke Qin, qinpanke@hpu.edu.cn

complexity and non-linearity of financial data make the prediction process full of challenges. Therefore, the study of prediction methods with high accuracy and universality has become a hot topic among scholars. At present, common forecasting methods for financial data include traditional econometric methods and machine learning methods.

The econometric method is based on statistical theories to predict the financial market. Commonly used models include Autoregressive Integrated Moving Average (ARIMA) (*Box & Jenkins, 1968*), Generalized Autoregressive Conditional Heteroskedasticity (GARCH) (*Sobreira & Louro, 2020*) and Vector Autoregressive (VAR) (*Kiss, Mazur & Nguyen, 2022*) models. Although the econometric model has relatively objective theoretical support, the strict basic assumptions make it challenging to achieve the desired predictive effect on real data.

The emergence of machine learning has ushered in a new era for financial sequence forecasting. Currently, researchers in related fields are primarily focusing on two aspects: the extraction of Alpha factors (*Shen et al., 2023*) and the optimization of predictive models.

Alpha factors are typically a combination of various underlying models that have been evaluated using historical datasets. They reflect the underlying drivers of unexplained asset price movements. Investors can potentially achieve excess returns by incorporating these alpha factors into their trading strategies. In addition, factor mining is also used for unstructured data such as news articles and social media posts (*Lin, Tsai & Chen, 2022*). They are able to capture market sentiment and underlying influences through their models, thereby enhancing the accuracy of forecasts.

In the field of model research, a primary task is predicting price data, which is a typical time series with characteristics such as multi-scale and high noise. The rich characteristics of price data make it a hot research field for time series forecasting (TSF), attracting a large number of researchers to challenge. There are four main kinds of methods for time series modeling: (i) Traditional machine learning algorithms. Traditional machine learning algorithms require less data and the decision-making process is easier to explain. However, the simple structure limits their ability to handle complex data. (ii) Recurrent neural networks (RNNs) (*Hochreiter & Schmidhuber, 1997*). RNN models learn the hidden states of the time series through recursion and have demonstrated high accuracy in certain time series prediction tasks. In addition, fast inference can be achieved due to the fact that the output is dependent solely on the current state and the input from the previous moment. However, this structure also makes it challenging to achieve efficient parallel training. (iii) Transformer-based models (*Vaswani et al., 2017*; *Li et al., 2019*; *Liu et al., 2021*; *Zhou et al., 2021*; *Wu et al., 2021*). The core of Transformer is the attention layer. The efficacy of self–attention mechanism is attributed to its ability to capture key information within a window, allowing it to model complex data. Although it demonstrates excellent performance in both multidimensional feature processing and long-term prediction work, the demand for the amount of training data and computational cost also presents a challenge to its practical application. (iv) Temporal convolutional networks (TCN) (*Borovykh, Bohte & Oosterlee, 2017*; *Bai, Kolter & Koltun, 2018*; *Liu et al., 2022a*). The TCN employs the dilated convolutions in order to more effectively identify long-term dependencies. Additionally, the

shared parameters of the convolutional layer enhance computational efficiency. However, this convolution process also results in diminished sensitivity to local information.

Although improvements in the structure of predictive models are crucial, a combination of different approaches, such as optimizing the hyperparameters of models (*Zrieq et al., 2022*; *Wang et al., 2022*; *Liu et al., 2022b*), employing feature selection techniques (*Wu et al., 2023*), and decomposing time-series characteristics (*Ke et al., 2023*), plays an important role in enhancing the performance of models. *Huang et al. (2023)* utilized the improved complementary ensemble empirical mode decomposition (ICEEMDAN) algorithm to decompose the nonferrous metal price series. The fuzzy entropy value is calculated to select suitable models for the sub-sequence, including ARIMA and gated recurrent unit (GRU) with Bayesian optimization. Experimental results show that hyperparameter optimization and hybrid frameworks can effectively improve prediction accuracy. *Wang, Zhuang & Gao (2023)* employed a hybrid system for predicting carbon price prediction. The method combines the strengths of several techniques, including the use of Singular Spectrum Analysis (SSA) to reduce noise and chaotic disturbance, Extreme Gradient Boosting (XGBoost) and partial autocorrelation function (PACF) to identify valid factors, and the Slime Mold Algorithm (SMA) to ensure the accuracy and stability of the overall system. Furthermore, *Ashrafzadeh et al. (2023)* combined particle swarm optimization (PSO) and the K-means method to make convolutional neural network (CNN) perform on stock prediction without any significant difference from traditional methods. These demonstrate that predictive models can be improved by leveraging the strengths of other methods without adding complexity. Among these, feature selection techniques and temporal decomposition reduce the difficulty of model learning through certain preprocessing, while hyperparameter optimization adjusts the model's untrained parameters to enhance its ability to adapt and learn from the data.

The objective of this article is to enhance the performance of arbitrage spread prediction models, thereby improving arbitrage trading strategies. Although neural network models with superior predictive capabilities have made some progress in structural exploration and have achieved great results in long-term prediction and multi-feature processing, they also have some problems such as complex model structure, high usage costs, and difficult adjusting. Therefore, in the field of short-term forecasting of financial series, traditional neural networks are still widely used due to their simpler model structure, less difficulty in training and excellent fitting effect. Among these, numerous scholars (*Sheng & Ma, 2022*; *Zhan et al., 2022*) have demonstrated that long short-term memory (LSTM) networks perform better in several financial sequence prediction tasks. However, most models suffer from degraded predictive performance in cross-data migration applications due to limitations in model size and data volume. This is more pronounced in traditional neural networks. The performance of a model can be improved by adjusting some non-training parameters of the model, namely hyperparameter tuning. Compared to grid search, relying on the researcher's choice to improve traversal efficiency, random search based on swarm intelligence algorithms can continuously exploit population behaviour for comprehensive search during the optimization process, and therefore has good search capability in solving hyperparameter optimization problems (*Liu et al., 2022b*;

*Ashrafzadeh et al., 2023*). Therefore, this article constructs a nested two-layer optimization network to learn hyperparameters using swarm intelligence algorithms and parameters using gradient descent. This approach enables automatic tuning of hyperparameters and improves prediction performance in cross-data model migration.

Specifically, this article conducts research on LSTM network to predict future arbitrage spread movements. Compared to complex neural network models, the LSTM network structure is relatively simple and intuitive to comprehend. But model's performance is more susceptible to the chosen hyperparameters. Based on this, in order to get better global search results, we propose a novel hyperparameter optimization algorithm, namely K-fold Cross-Search (KCS), to learn the traditional non-training parameters and improve the accuracy of arbitrage spread predictions. The main contributions of this study can be summarized as follows:

- We propose a novel optimized algorithm (KCS) which integrates the advantages of grid search and ant pheromone mechanisms with enhanced global search capability.
- We constructed the KCS-LSTM prediction network. It has been implemented for predicting arbitrage spreads in real-world Rebar (RB) and Hot-rolled coils (HC) futures contracts.
- We compare the predictive performance of KCS-LSTM with some deep learning modes including Back Propagation neural network (BPNN), RNN, Long- and Short-term Time-series network (LSTNet), Informer and Sample Convolution and Interaction network (SCINet), the results show that KCS-LSTM has achieved great success.

The remainder of this article is organized as follows: 'Data Preparation' describes the data preparation in the arbitrage spread application, including the construction of the objective function, the acquisition of raw data, the construction of the spread factor, and the cointegration analysis on the arbitrage portfolio. 'Methodology' introduces the KCS algorithm and the KCS-LSTM network. 'Experiment' presents the experimental description and result analysis of KCS-LSTM and other models for intercommodity spread prediction. 'Conclusion' presents the conclusion of this article.

## DATA PREPARATION

In this section, we put forward the primary objective function. Additionally, we provide a brief description of the data preparation and affirm its validity and applicability through cointegration analysis.

### Mathematical modeling

Our objective is to develop an arbitrage spread prediction model with higher accuracy. To validate the efficacy of the proposed methodology, historical data related to RB and HC futures contracts traded on the Shanghai Futures Exchange are used to simulate spread prediction in cross-species arbitrage trading. Specifically, we assume that the closing spread of the arbitrage portfolio in the subsequent minute serves as a guide for executing the arbitrage trading strategy. The LSTM model will be trained using a subset of the spread data to make predictions on new test data. The test set is not allowed to be involved in

any calculations related to the training process. At this point, the original objective is transformed into the minimization of the test set loss function, and the main objective function can be expressed as:

$$minimize_{\Theta} \sum_{t \in \Omega_{test}} \| y_t - model_{\Theta}(X_t) \|_F^2. \tag{1}$$

where $\Theta$ denotes the hyperparameter set of our model, $\Omega_{test}$ is the set of time stamps used for testing, $F$ is the Frobenius norm, $model_{\Theta}$ is the predictive model, $X_t$ encompasses all the feature data within the observable range when making predictions about $y_t$ and $y_t$ is the real value of the closing price spread at time $t$.

## Data description

Our data is sourced from the Shanghai Futures Exchange in China. It provides a snapshot-based order feed using the CTP protocol. The order feed aggregates changes over the last 500 ms. We use 500-millisecond tick data for the rebar and hot-rolled coil contracts to calculate the spread data. The final output is 1-minute K-line data. In addition, since each contract typically lasts for one year, we spliced the historical data for the January, May, and October contracts of each year based on turnover to obtain the continuous spread data of the main contract. In the end, we obtained the spread data from 21:01 on 15 July 2020 to 10:50 on 23 March 2023, comprising a total of 225,155 data points spanning 654 days. Each data point corresponds to the spread change within 1 min, including the following eight characteristics:

1. OPEN/HIGH/LOW/CLOSE: the first/highest/lowest/last value in 1-minute spread data.
2. Difference (DIF): $DIF_i = EMA(CLOSE, 12) - EMA(CLOSE, 26)$. (EMA is the exponential moving average)
3. Differential Exponential Average (DEA): $DEA_i = EMA(DIF_i, 9)$.
4. Moving average convergence and divergence (MACD): $MACD_i = 2 \times (DIF_i - DEA_i)$.
5. The price spread fluctuation.

## Cointegration analysis

Before initiating intercommodity spread trading, it is crucial to confirm the existence of a long-term stable cointegration relationship among the selected futures contracts. To facilitate this, we employed EViews10 software for the cointegration analysis of the original price data.

Upon examining the contract time series plot depicted in Fig. 1, it becomes evident that the closing price data for both RB and HC exhibit similar fluctuation patterns. This initial observation suggests a potential correlation between the price data of these two commodity futures. A more detailed, quantitative analysis of this correlation is provided in Table 1. The correlation coefficients calculated for the opening price, closing price, highest price, and lowest price all point towards a significant correlation between the two commodities.

The stationarity test results presented in Table 2 reveal that for the price series of RB and HC, the null hypothesis of a unit root cannot be rejected at the 5% confidence level.

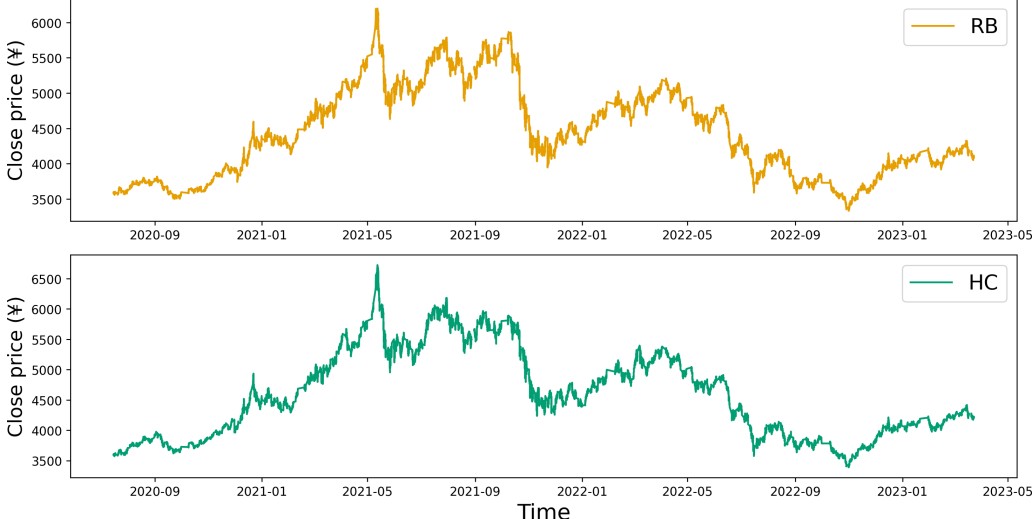

**Figure 1  Time series plot of the closing prices for RB and HC.**

**Table 1  Correlation test of the time series.**

| CORR | Open | Close | High | Low |
|---|---|---|---|---|
| RB-HC | 0.990999 | 0.991001 | 0.991005 | 0.990997 |

It suggests that all eight series are non-stationary. Upon applying first-order differencing, the calculated Augmented Dickey-Fuller (ADF) test statistic is less than the critical value. It provides a basis for rejecting the null hypothesis of a unit root, indicating that all first-order difference series are stationary. Consequently, it can be inferred that the price series of the main futures contracts of both RB and HC are integrated of order 1.

Next, we can proceed with the Engle-Granger cointegration test, which starts with formulating the following cointegration equation:

$$hc\_close = c \cdot rb\_close + et \tag{2}$$

where $et$ is referred to as a cointegration residual, or a residual for short. The parameter $c$ is known as a cointegration coefficient.

Table 3 shows the results of the EG cointegration test. At the 1% confidence level, the ADF test statistic of the residual series is lower than the critical value. Thereby, we reject the null hypothesis and deem the series as stationary. Consequently, following the EG cointegration theory, it can be inferred that the price data for the main contracts of RB and HC adhere to a cointegration relationship. We can conduct pair trading on this data.

To validate the effectiveness of our fitting process, we conducted a stationarity test on the fitted spread data. As shown in Table 4, the fitted spread data are the stationary time series at the 5% confidence level. It suggests that our fitting process is robust and reliable. The time series plot of the fitted data for the closing price spread is shown in Fig. 2.

**Table 2 ADF test on the price data of RB and HC.**

| | Series | ADF test statistic | Critical value | | | P-value | Stationarity (5%) |
|---|---|---|---|---|---|---|---|
| | | | 1% | 5% | 10% | | |
| RB | CLOSE | −0.045 | | | | 0.6678 | |
| | OPEN | −0.077 | −2.565 | −1.941 | −1.617 | 0.6571 | no |
| | HIGH | −0.067 | | | | 0.6605 | |
| | LOW | −0.052 | | | | 0.6656 | |
| HC | CLOSE | −0.033 | | | | 0.6719 | |
| | OPEN | −0.077 | −2.565 | −1.941 | −1.617 | 0.6571 | no |
| | HIGH | −0.062 | | | | 0.6621 | |
| | LOW | −0.061 | | | | 0.6625 | |
| DrB | CLOSE | −344.521 | | | | 0.0001 | |
| | OPEN | −342.534 | −2.565 | −1.941 | −1.617 | 0.0001 | yes |
| | HIGH | −325.768 | | | | 0.0001 | |
| | LOW | −326.774 | | | | 0.0001 | |
| | CLOSE | −345.716 | | | | 0.0001 | |
| DHC | OPEN | −490.118 | −2.565 | −1.941 | −1.617 | 0.0001 | yes |
| | HIGH | −326.216 | | | | 0.0001 | |
| | LOW | −208.269 | | | | 0.0001 | |

**Table 3 ADF test of the residual series.**

| Residual | ADF test statistic | Critical value | | | P-value | Conclusion |
|---|---|---|---|---|---|---|
| | | 1% | 5% | 10% | | |
| $e_t$ | −4.52 | −2.57 | −1.94 | −1.61 | 0.0001 | cointegration |

**Table 4 ADF test of the fitted spread series.**

| Series | ADF test statistic | Critical value | | | P-value | Stationarity (5%) |
|---|---|---|---|---|---|---|
| | | 1% | 5% | 10% | | |
| CLOSE | −2.054 | | | | 0.0384 | |
| OPEN | −2.097 | −2.565 | −1.941 | −1.617 | 0.0346 | yes |
| HIGH | −2.088 | | | | 0.0354 | |
| LOW | −2.0829 | | | | 0.0358 | |

# METHODOLOGY

The KCS algorithm is designed to determine the optimal hyperparameters in this section. Further, we propose the KCS-LSTM network to adaptively tune the prediction model. The network eliminates the impact of individual subjectivity elements in traditional

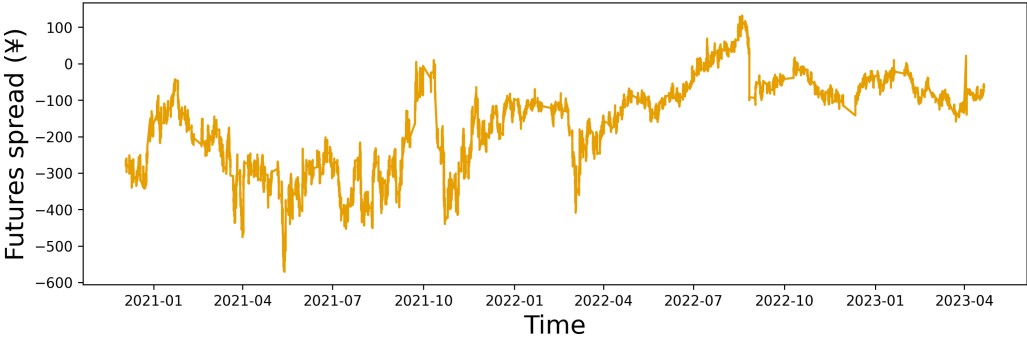

**Figure 2** **Time series plot of the closing price spread.**

hyperparameter tuning and offers a reliable reference for constructing effective and efficient predictive models in the field of algorithmic trading.

## K-fold cross search
### Related work

Hyperparameter tuning is the process of finding the optimal combination of hyperparameters that maximizes the output accuracy. There are several methods for hyperparameter tuning in artificial neural network (ANN), such as grid search (*Bergstra & Bengio, 2012*), Bayesian optimization (*Snoek, Larochelle & Adams, 2012*), gradient-based optimization (*Maclaurin, Duvenaud & Adams, 2015*), or random search based on metaheuristic algorithms (MAs).

Traditional grid search is a common method used for hyperparameter tuning in the development of time series forecasting models. However, it can pose some challenges. If the grid points are too sparse, the optimal search outcome may deviate significantly from the true optimal solution. Conversely, a densely populated grid could significantly increase the computational demands of the experiment. Therefore, selecting the appropriate grid granularity and extent is critical to maintaining a balance between grid density and computational load. However, individual subjective factors often influence the process of achieving this balance.

For the deficiencies of traditional grid search, population-based MAs have been widely used for combinatorial optimization problems. These algorithms generate optimal solutions by exploring a new region in the search space *via* an iterative process. Grey Wolf Optimizer (*Mirjalili, Mirjalili & Lewis, 2014*), Particle Swarm Optimization (*Kennedy & Eberhart, 1995*), Cuckoo Search Algorithm (*Gandomi, Yang & Alavi, 2013*), and Sparrow Search Algorithm (*Xue & Shen, 2020*) are well-known examples of this class of MAs. The Ant Colony Optimization (ACO) is a population-based metaheuristic algorithm proposed by *Dorigo, Maniezzo & Colorni (1996)*. It emulates the behavior of ant colonies as they seek the shortest path during their food-foraging process and has parallel computing, strong robustness, and easy combination with other algorithms. To better handle complex optimization problems, the ACO algorithm has been enhanced in three main areas. Firstly, the search, update, and coordination mechanisms have been improved in the feedback

mechanism of pheromones. *Engin & Güçlü (2018)* proposed a crossover and mutation mechanism to solve shop scheduling. *Zhao, Zhang & Zhang (2020)* presented an adaptive reference point mechanism to enhance the optimization ability. Secondly, to address the shortcomings of the algorithms themselves, a hybrid ACO with other algorithms was proposed. *Wang & Han (2021)* presented a hybrid algorithm with symbiotic organism search, ACO, and local optimization strategy. Finally, some parameters of the ACO algorithm were adaptively tuned. *Yang et al. (2016)* developed an adaptive parameter adjustment by taking the differences among niches into consideration, and a differential evolution mutation operator for ants.

Neural network hyperparameter tuning can be considered a black-box optimization problem. When applying ACO to black-box optimization problems, the difficulty in evaluating the search space makes the selection of the population size of the ACO algorithm a challenge. Additionally, the revised method of continuously exploring new areas encounters challenges in determining the appropriate number of iterations due to the heightened randomness of the global search. This article introduces a novel method of hyperparameter tuning based on the principles of the traditional ant colony algorithm. It allows for the automatic adjustment of hyperparameters in ANN models across a wide range of options. In our algorithm, the basic unit of the search space is an interval block instead of a coordinate point. We limit the total number of basic units in the searchable space to ensure that the difficulty of the search process does not outweigh the capabilities of the pheromone search mechanism. During the iterative process, each search unit eliminates unpromising regions using the worst elimination rule. The size of the interval block decreases as the search progresses until it reaches a critical point, indicating the end of the search. The critical point here is determined by the precision of the hyperparameters. If the size of the region eliminated in each interval block update is fixed, the number of iterations also needs to increase. Therefore, the demand for the number of populations in the original algorithm has been replaced with a demand for the number of iterations. Our approach utilizes the reduction process of the interval block as a reference, making it easier to select the number of iterations. More details about the search space and search rules will be elaborated in 'Search space' and 'Search rules'.

### Search space

The KCS algorithm uses a transformation mechanism to convert the search space into a search environment where interval blocks serve as the basic unit. Since the size of $K$ and the number of hyperparameters $m$ are fixed values, the original search space of any size will be transformed into a new search space of a finite size in each iteration. The performance of the search algorithm is only affected by the distribution of the new space. Here the new search environment will be explored by ACO algorithm. Unpromising regions are excluded, and the remaining regions will again generate a new search environment. As the iterative process proceeds, the size of the basic unit of the search environment will continue to shrink and the average fitness level of the entire search environment will be optimised.

The *Split-Range1* algorithm and *Split-Range2* algorithm will perform the conversion process. *Split-Range1* will handle initialization and spatial updates under the Optimal

Selection Rule, while *Split-Range2* will handle spatial updates under the Worst Elimination Rule. The basic search unit in the new search environment is denoted as $B = \{L_{1j_1}, L_{2j_2}, \ldots, L_{mj_m} | j_i \in 1, 2, \ldots, K\}$ and referred to as an interval block. The *Split-Range1(2)* algorithm divides the range of values $L_i(i = 1, 2, \ldots, m)$ of each hyperparameter into $K$ parts, creating $m \times K$ subintervals, which make up the $K^m$ basic units of the new search environment.

The conversion process of the *Split-Range1(2)* algorithm is shown in Algorithm 1.

---

**Algorithm 1:*Split-Range*1**

---

| | |
|---|---|
| 1: | **Input:** $b.list$      % the sequence number set of intervals |
| | $b[low/high, length(b.list)]$ % boundary of intervals |
| 2: | **Description:** |
| | $bd[low/high, m, K]$ % the boundary matrix of the sub-intervals |
| | $bp[m, K]$ % the representative value matrix of the sub-intervals |
| 3: | $diff \leftarrow (b[high] - b[low])/K$ |
| 4: | **for** $i \leftarrow 0 : K - 1$ **do** |
| 5: | $bd[low, b.list, i] \leftarrow i \cdot diff + b[low]$ |
| 6: | $bd[high, b.list, i] \leftarrow (i+1) \cdot diff + b[low]$ |
| 7: | **end for** |
| 8: | $bp[b.list, 0 : K - 1] \leftarrow round((bd[high] - bd[low])/2)$ |
| | % The *round* function performs rounding to the nearest precision |
| 9: | **Output:** $bp, bd$ |

---

***Search rules***

Assuming that the initial pheromone concentration of each subinterval is $\tau(0)$, and the number of ants searching the interval blocks is $n$. The optimization process in each iteration primarily relies on the principles of the ant colony algorithm and involves the following three types of updates:

(1) Update of the pheromone matrix

The pheromone matrix serves as a comprehensive record of search information across the entire search space. It is influenced by both positive and negative feedback mechanisms of pheromones.

In the case of positive feedback, whenever an ant completes a search of an interval block, it releases a corresponding amount of pheromones according to Eq. (3). The amount of pheromones released reflects the quality of the ants' search results.

$$\Delta\tau_{ij} = \sum_{k=1}^{n} \frac{Q \cdot IS_{ij}(k)}{Fit_k}, (k = 1, 2, \ldots, n) \tag{3}$$

where $\Delta\tau_{ij}$ represents the pheromone increment of the $j$-th subinterval for the $i$-th hyperparameter, the constant $Q$ is the adjustment factor of the pheromone increment, and

**Algorithm 2:*Split-Range*2**

| | |
|---|---|
| 1: | **Input:** $b.list$      % the sequence number set of intervals |
| |      $b[low/high, length(b.list), K-1]$      % boundary of intervals |
| 2: | **for** $t \leftarrow 0 : length(b.list) - 1$ **do** |
| 3: |      Find the independent intervals $T$ from $b[\{low, high\}, t, \{0 : K-1\}]$ |
| |              % independent intervals are not adjacent to each other |
| 4: |      Count the number $T.num$ and the size $T.size$ of $T$. |
| 5: |      $g[T.num]$ |
| 6: |      $g \leftarrow round(K \cdot T.size / sum(T.size))$ |
| 7: |      **if** $sum(g) \neq K$ **then** |
| 8: |          **for** $i \leftarrow 0 : T.num - 1$ **do** |
| 9: |              **if** $T.size[i] \geq Sort(T.size)[K \bmod T.num]$ **then** |
| 10: |                  $g[i] \leftarrow K//T.num + 1$ |
| 11: |              **else** |
| 12: |                  $g[i] \leftarrow K//T.num$ |
| 13: |              **end if** |
| 14: |          **end for** |
| 15: |      **end if** |
| 16: |      $T.size[i]$ is divided into $g[i]$ *intervals.* |
| |      Returning the bounds $[bs.low, bs.high]$ of each interval |
| 17: |      $bd[low, b.list[t]] \leftarrow bs.low$ |
| 18: |      $bd[high, b.list[t]] \leftarrow bs.high$ |
| 19: | **end for** |
| 20: | $bp[b.list, 0 : K-1] \leftarrow round((bd[high] - bd[low])/2)$ |
| 21: | **Output:** $bp, bd$ |

$Fit_k$ is the quantified indicator of the search results of the $k$- th ant. When $L_{ij}$ is a part of the interval block searched by the $k$ th ant, $IS_{ij}(k) = 1$, otherwise $IS_{ij}(k) = 0$.

In the case of negative feedback, the pheromones on each subinterval will evaporate over time.

The specific update formula for the pheromone matrix is as follows:

$$\tau_{ij}(t+1) = (1-\rho)\tau_{ij}(t) + \Delta\tau_{ij}, (0 < \rho < 1) \tag{4}$$

where $\rho$ is the evaporation factor, and $\tau_{ij}(t)$ is the pheromone concentration on $L_{ij}$ during the $t$- th iteration.

(2) Update of the selection probability matrix

The selection probability matrix is updated along with the pheromone matrix, and the calculation formula is as follows:

$$p_{ij}(t) = \frac{\left(\tau_{ij}(t)\right)^{\alpha}}{\sum_{j=1}^{K}\left(\tau_{ij}(t)\right)^{\alpha}}, (i = 1, 2, \ldots, m) \tag{5}$$

where $p_{ij}(t)$ signifies the likelihood of a subinterval being chosen in the $t$- th iteration. The sensitivity of the ants is modulated by $\alpha\left(\alpha \in N^{+}\right)$, known as the sensitivity factor. A larger value of the sensitivity factor indicates a heightened sensitivity of the ants to variations

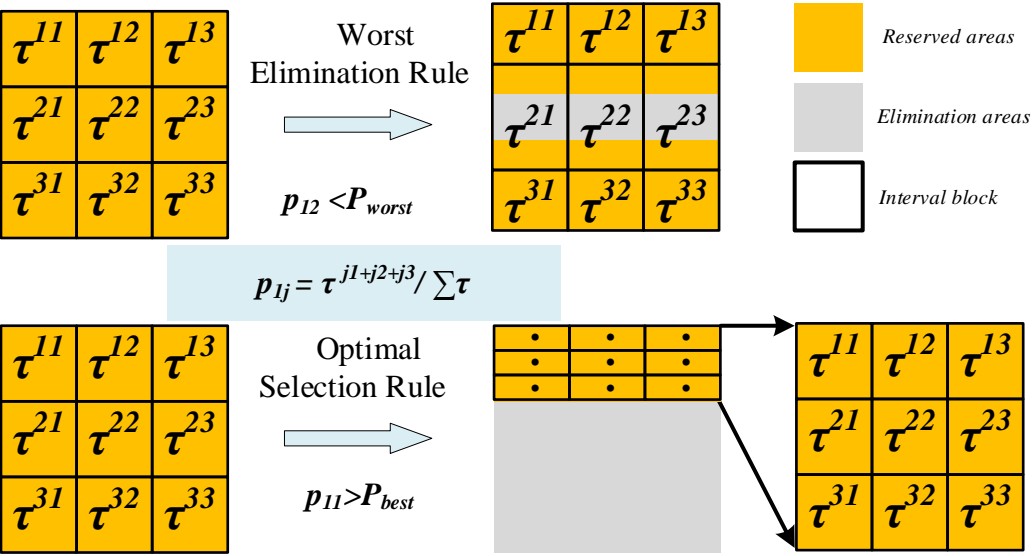

**Figure 3** Update of the search space ($m = 2$, $K = 3$).

in pheromone concentration. Based on the selection probability matrix, we choose the interval block that each ant will search in the next iteration by roulette.

(3) Update of the search space

The update of the search space involves two mechanisms (as shown in Fig. 3): optimal selection and worst elimination.

Optimal selection rule: When the selection probability of a subinterval exceeds the threshold $P_{best}$, it will become the only available value range for the corresponding hyperparameter and forms a new search space with the value ranges of other hyperparameters.

Worst elimination rule: When the selection probability of a subinterval A is less than the threshold $P_{worst}$, if the value range of subinterval A is adjacent to the neighboring subinterval B, we retain a quarter of the interval length of A that is close to B, otherwise, discard subinterval A entirely. Note that the selection probability of A is the minimum of $K$ probability values for a hyperparameter.

Based on the above search rules, the search process of the KCS algorithm is shown in Algorithm 3.

## KCS-LSTM network

Figure 4 illustrates the architecture of the KCS-LSTM model. Each subheading in the third part of the article corresponds to a specific part of the chart. Our KCS-LSTM network is structured with two nested layers. The inner layer, which consists of an LSTM network, uses the hyperparameters provided by the outer layer to initialize the model and trains the model's parameters *via* gradient descent. Subsequently, it relays the corresponding fitness function values back to the outer layer. The outer layer, a KCS hyperparameter optimization network, iteratively searches for the optimal hyperparameter set using a limited number of

**Algorithm 3: KCS**

| | |
|---|---|
| 1: | **Input:** $m$, $n$, $L[m, 2]$, $item$ |
| 2: | Initialize *the* $bd[2, m, K]$, $bp[m, K]$, $\tau[m, K]$. |
| 3: | *Split Range* $1(L, [0, 1, \ldots, m])$. |
| 4: | **for** $gen \leftarrow 1 : item$ **do** |
| 5: | Calculate the matrix $p[m, K]$ %Eq. (5) |
| 6: | **if** $\exists p[i, j] > P_{best}$ **then** |
| 7: | $b[2, length(\{i | p[i, j] > P_{best}\})], b \leftarrow bd[0 : 1, i, j]$ |
| 8: | $b.list \leftarrow \{i | p[i, j] > P_{best}\}$ |
| 9: | *Split Range* $1(b, b.list)$ % Algorithm 1 |
| 10: | Update $\tau[b.list, 0 : K-1]$ and $p[b.list, 0 : K-1]$ |
| 11: | **end if** |
| 12: | **if** $\exists \min(p[i, ]) < P_{wrost}$ **then** |
| 13: | Worst Elimination Rule |
| 14: | $b[2, length(\{i | \min(p[i, ]) < P_{wrost}\}), K-1]$ |
| 15: | $b \leftarrow bd[0 : 1, i, \{0, 1, \cdots, j-1, j+1, \cdots, K-1\}]$ |
| 16: | $b.list \leftarrow \{i | \min(p[i, ]) < P_{wrost}\}$ |
| 17: | *Split Range* $2(b, b.list)$ % Algorithm 2 |
| 18: | Update $\tau[b.list, 0 : K-1]$ and $p[b.list, 0 : K-1]$ |
| 19: | **end if** |
| 20: | Based on $p$ and roulette , calculate $Ak$ %Eq. (12) |
| 21: | $Pi \leftarrow bp \cdot Ai$ |
| 22: | Calculate the fitness $Fiti$ in $Pi$ |
| 23: | Update $\tau$ %Eqs. (3)–(4) |
| 24: | **end for** |
| 25: | **Output:** Values of $bp[0 : m, \{j | \tau[i, j] = \max(\tau[i, ])\}]$ |

specified sample points. The operational process of the entire KCS-LSTM network will be elaborated in 'Steps of the KCS-LSTM'.

### *LSTM network*

Our LSTM network is depicted in Fig. 4. It comprises an LSTM layer and a fully connected (FC) network. The number of neurons in each layer serves as a hyperparameter and is determined through the KCS optimization algorithm. Each LSTM unit is built using the traditional 'gate' structure. Their internal mechanism is described by the following equations:

$$\text{Input Gate}: \quad I_t = sigmoid\left(w_i \cdot \left[d_t, h_{t-1}\right] + b_i\right) \tag{6}$$

$$\text{Output Gate}: \quad O_t = sigmoid\left(w_o \cdot \left[d_t, h_{t-1}\right] + b_o\right) \tag{7}$$

$$\text{Forget Gate}: \quad f_t = sigmoid\left(w_f \cdot \left[d_t, h_{t-1}\right] + b_f\right) \tag{8}$$

$$\text{Cell state}: \quad C_t = f_t \cdot C_{t-1} + I_t \cdot \tanh\left(w_c \cdot \left[d_t, h_{t-1}\right] + b_c\right) \tag{9}$$

$$h_t = O_t \cdot \tanh(C_t) \tag{10}$$

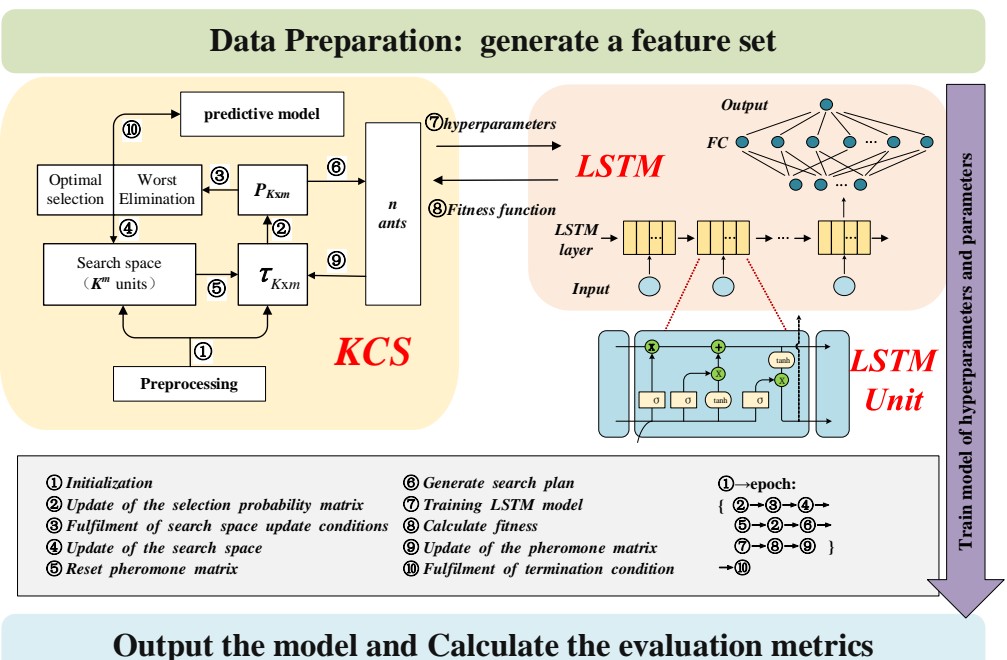

**Figure 4** KCS-LSTM network.

where weight matrixes $(w_i, w_o, w_f, w_c)$ and bias vectors $(b_i, b_o, b_f, b_c)$ of $I_t, O_t, f_t$ and $C_t$ are obtained through a process of training on the designated training set. $h_t$ represents the extracted feature of each LSTM unit and will be used as the input feature for the next time step. Note that only the $h_t$ from the final time step is utilized as the extracted feature of the LSTM layer. This feature is subsequently input into a straightforward FC network and transformed into the predicted value $(\hat{y}_t)$.

### *Fitness function*

In numerous models for hyperparameter optimization, the mean squared error (MSE) or mean absolute error (MAE) of the validation set is typically employed as the fitness function of the model. However, to mitigate the risk of model overfitting, we design a novel fitness function:

$$Fit = \frac{1}{n-start} \sum_{i=start}^{n} Loss\_fun_{val}^{i} \cdot \left| Loss\_fun_{val}^{i} - Loss\_fun_{train}^{i} \right| \tag{11}$$

where $Loss\_fun_{val}^{i}$ is the loss function value of the model on the validation set after training for $i$ epochs. The efficacy of the newly proposed fitness function will be substantiated through the experimental results presented in 'Experiment'.

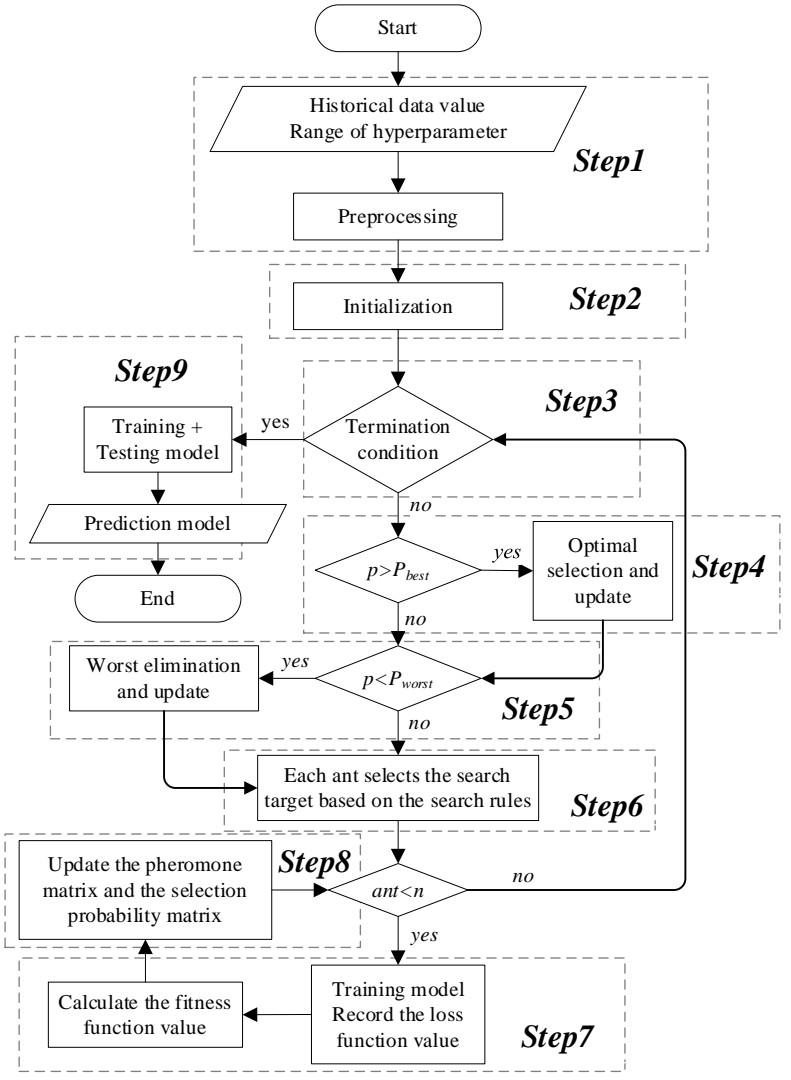

**Figure 5 Flowchart of the KCS-LSTM network.**

### Steps of the KCS-LSTM

The flowchart of the KCS-optimized LSTM neural network is shown in Fig. 5. The main steps are as follows:

*Step 1*. Preprocessing. Preprocess the historical data, including completing the division of training set, validation set, and test set; eliminate the dimensional differences between data through normalization.

    *Step 2*. Initialization. Set the number of populations, K value, number of algorithm iterations, *etc*. Then, initialize the search space and the pheromone matrix.

    *Step 3*. Termination condition. The termination condition is defined as follows:

(1). The process ends when the maximum number of iterations is reached.

(2). It also ends when the early termination criteria are met. This occurs when the size of the subinterval is less than or equal to the precision of its corresponding hyperparameter.

At these points, the KCS optimization process concludes. The representative value of the interval block exhibiting the highest pheromone concentration is then returned as the final result of the hyperparameter optimization. Subsequently, the process advances to Step 9. If not, continue with the KCS optimization.

*Step 4.* Optimal selection. The optimal selection rule is implemented. The objects of conditional judgment do not include probability values where the size of the corresponding subinterval is less than or equal to the precision of the hyperparameter. Finally, it returns the updated search space, pheromone matrix, and selection probability matrix.

*Step 5.* Worst elimination. The worst elimination rule is implemented. The objects of conditional judgment do not include probability values where the size of the corresponding subinterval is less than or equal to the precision of the hyperparameter. Finally, it returns the updated search space, pheromone matrix, and selection probability matrix.

*Step 6.* Search plan. Based on the selection probability matrix and roulette, determine the interval blocks that each ant will search in this iteration. The search plan is represented as:

$$A_k = \left[ IS_{ij}(k) \right]_{m \times K}, \left( i = 1, 2, \ldots, m; j = 1, 2, \ldots, K; k = 1, 2, \ldots, n \right). \tag{12}$$

Moreover, a set of representative values is calculated for the interval blocks to be searched. In this article, the median of the subinterval is used as the representative value. These values serve as hyperparameters, playing a crucial role in both the initialization and the training processes of the model.

*Step 7.* Calculate fitness. Under the given hyperparameters, the sliding window mechanism is employed to extract the feature data and the label data, represented as:

$$D_{feature} = \left\{ d_1, d_2, \ldots, d_t, \ldots, d_{L_{in}} | d_t = x_{t1}, x_{t2}, \ldots, x_{tM_{in}} \right\},$$
$$D_{targets} = \left\{ d_{L_{in}+1} = x_{t'i1}, x_{t'i2}, \ldots, x_{t'iM_{out}} | i^j \in \{1, 2, \ldots, M_{in}\}, t' = L_{in} + 1 \right\}$$

where $L_{in}$ is the length of the input window for the time series, $M_{in}$ is the number of input features, and $M_{out}$ is the number of features to be predicted. $D_{feature}$ are substituted into Equations (6)–(10). The extracted features are then transformed into predicted values using a fully connected network. Next, the loss function is calculated, and the model parameters are updated through backpropagation. Finally, the loss function values of the training set and validation set throughout the entire training process are substituted into Eq. (11). We obtain the fitness corresponding to the given hyperparameters.

*Step 8.* Update. The pheromone matrix is updated according to Eqs. (3) and (4). Subsequently, the selection probability matrix is recalculated using Eq. (5). The process then returns to step 3.

*Step 9.* Prediction model. The hyperparameters obtained from step 3 are employed to define and train the neural network model. Subsequently, the final predictive model is generated.

**Table 5  Experimental environment.**

| Item | Type |
|------|------|
| Operating System | CentOS 7.9.2009 |
| CPU | Intel(R) Xeon(R) Bronze 3204 @ 1.90 GHz |
| GPU | NVIDIA Tesla V100 (32G) |
| Deep Learning Frame | Pytorch 2.0.0 |
| Acceleration Library | CUDA 11.7+cuDNN 8.9.0 |

# EXPERIMENT

This section delves into the experimental aspect of the proposed model. 'Experimental Details' offers a detailed explanation of the experimental environment, parameter configurations, and the results of optimization, *etc*. 'Results and Analyses' provides a comprehensive description and analysis of the comparative experimental outcomes.

## Experimental details
### Experimental environment

All the models were trained/tested in the same experimental conditions, and detailed configuration is given in Table 5.

### The processing of data

We have selected 1-minute cycle K-line fitting price spread data as the experimental data to test the performance of the proposed forecasting model. And the price spread data for each period (1 min) consists of eight features, including the opening price spread, the highest price spread, the lowest price spread, the closing price spread, MACD, DEA, DIF, and the price spread fluctuation (See 'Data description'). Among these features, the closing price spread is our target for prediction. In all experiments, the datasets (190,000 min) have been split into a training set (128,000 min), a validation set (30,000 min) and a test set (32,000 min) in chronological order.

Each data is normalized to the range of 0 to 1 when the characteristic data is input into the artificial neural network. It reduces the effects of noise, ensuring that neural networks update parameters efficiently and speed up the training of the network. We use the following formula for normalization.

$$\overline{x}(t) = \frac{x(t) - x_{min}}{x_{max} - x_{min}} \tag{13}$$

where $x_{min}$ and $x_{max}$ are the minimum and maximum values of each feature in the training set respectively. Since the data is normalized during the model training phase, the output of the test set can be restored by the formula $x(t) = x(t)'(x_{max} - x_{min}) + x_{min}$, where $x(t)'$ is the output value of the forecasting model.

### Metrics

Four error measures are adopted to assess the performance of the proposed models with more accuracy, including mean square error (MSE), symmetric mean absolute percentage

error (sMAPE) and root relative square error (RSE). They are calculated as follows:

$$e_{MAE} = \frac{1}{n} \sum_{t=1}^{n} |\hat{y}_t - y_t| \tag{14}$$

$$e_{MSE} = \frac{1}{n} \sum_{t=1}^{n} (\hat{y}_t - y_t)^2 \tag{15}$$

$$e_{sMAPE} = \frac{100\%}{n} \sum_{t=1}^{n} \frac{|\hat{y}_t - y_t|}{(|y_t| + |\hat{y}_t|)/2} \tag{16}$$

$$e_{RSE} = \sqrt{\frac{\sum_{t=1}^{n} (y_t - \hat{y}_t)^2}{\sum_{t=1}^{n} (y_t - \overline{y})^2}}. \tag{17}$$

In the above formula, $y_t$ represents the original value of the moment $t$, $\hat{y}_t$ represents the predicted value of the moment $t$, and $n$ is the total number of test samples. If the values of MAE, MSE, sMAPE, and RSE are smaller, the deviation between the predicted value and the original value is also smaller.

Besides the aforementioned indicators, we utilized two conventional evaluation metrics defined as:

• Empirical Correlation Coefficient (CORR):

$$e_{corr} = \frac{\sum_{t=1}^{n} (\hat{y}_t - \overline{y})(y_t - \overline{y})}{\sqrt{\sum_{t=1}^{n} (\hat{y}_t - \overline{y})^2} \sqrt{\sum_{t=1}^{n} (y_t - \overline{y})^2}} \tag{18}$$

which is used to test the degree of association between two variables,

• Symbol Accuracy (SA):

$$e_{SA} = \frac{100\%}{n-1} \sum_{t=1}^{n-1} z_t \tag{19}$$

where $z_t = \begin{cases} 1, & if\ (y_{t+1} - y_t)(\hat{y}_{t+1} - y_t) > 0 \\ 0, & if\ (y_{t+1} - y_t)(\hat{y}_{t+1} - y_t) < 0 \end{cases}, \overline{y} = \frac{1}{n} \sum_{t=1}^{n} y_t, \overline{\hat{y}} = \frac{1}{n} \sum_{t=1}^{n} \hat{y}_t$ . For both CORR and SA, a higher value indicates better performance.

### Model construction and hyperparameter tuning

All of our experiments use the Adam optimizer to optimize the parameters of model. The mean square error (MSE) is chosen as the loss function of model, and the batch size is 512.

In our KCS-LSTM network, the LSTM model includes an LSTM layer and a fully connected layer, beyond the input and output layers. To ensure a solid fit between the LSTM model and our data, the KCS-LSTM network optimizes the hyperparameters of the LSTM model using the KCS algorithm. We analyzed some research article (*Greff et al., 2016*; *Ding & Qin, 2020*) on the selection of hyperparameters for the LSTM model. The learning rate and the number of parameters in the model are crucial hyperparameters, as confirmed by our experimental results. Therefore, we ultimately decided to select five hyperparameters as optimization targets, including the initial learning rate, the number of neurons in each hidden layer (LSTM and FC), the time step of the input window, and epoch.

**Table 6   The details of KCS-LSTM network.**

| Parameters | Values |
|---|---|
| The number of ants | 20/10[*] |
| K | 5 |
| $P_{best}$ | 0.9 |
| $P_{worst}$ | 0.01 |
| $\alpha$ | 2 |
| The number of iterations | 50 |

**Notes.**
  *When the search space changes, the number of ants is 20, otherwise it is 10.

In addition, considering the constraints of the experimental conditions, we have defined a suitable range for hyperparameter search: the learning rate is between [0.001,0.01], the number of neurons in both the LSTM and Fully Connected layers is between [1,150], the time step is between [2,60], and the epoch is between [10,100]. More details of network components are provided in Table 6.

Figure 6 provides a visualization of the search space alterations throughout the iterative process. The $x$-axis represents the iteration count, with the current hyperparameter value range being recorded every fifth iteration. The $y$-axis, represented by the shaded region, corresponds to the feasible value interval. Table 7 provides a quantitative representation of the aforementioned content, with data recorded every tenth iteration. As depicted in Fig. 6 and Table 7, the optimization process for the search space is primarily driven by the worst elimination rule, with the optimal selection rule playing a secondary role in inducing changes. It can effectively mitigate the potential negative impact on search results that could arise from the weak representativeness of interval representative values.

Figure 7 illustrates the changing distribution of fitness values for the sample points throughout the iterative process. Each subplot's caption indicates the iteration count and the number of sample points included. Note that each iteration consists of 20 sample points. Fitness values that exceed the range of the $x$-axis are not shown in the figure. Furthermore, to facilitate representation, a logarithmic scaling transformation has been applied to the fitness values. As illustrated in Fig. 7, in the first iteration of the random search, the fitness of the 20 sample points has only 70% of the sample values within $[-3.3, -2.7]$, whose distribution center is around $-3$. The remaining 30% of the sample values are higher than $-2.7$. As the search process progresses, the center of the fitness distribution of the sample points, obtained by semi-random sampling relying on pheromones, shifts towards a smaller value of $-3.2$. The distribution transitions from being sparse to becoming more concentrated. It demonstrates the KCS algorithm's effectiveness in optimizing the search space.

Table 8 shows the optimal hyperparameters obtained for the KCS-LSTM network.

## Results and analyses

This part validates the effectiveness of the novel fitness function. And we conducted a comparative analysis of the predictive performance between KCS+LSTM, which employs the MSE value of the validation set as its fitness function, and KCS-LSTM, whose fitness

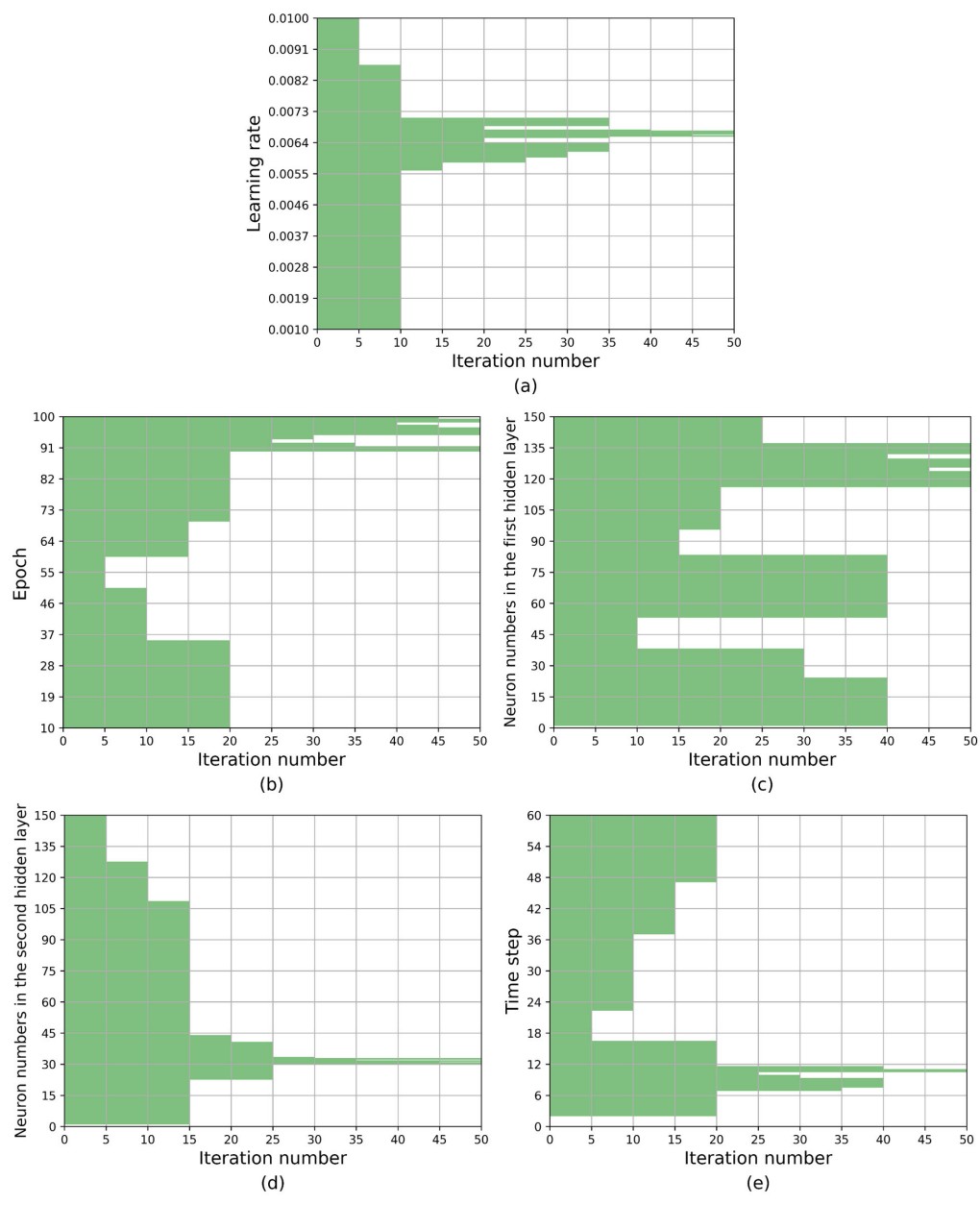

**Figure 6    The optimization process of the search space based on KCS-LSTM algorithm.**

function is computed as described in ' Fitness function'. It is worth noting that the KCS+LSTM network incorporates two dropout layers to mitigate overfitting. From Table 9 and Fig. 8, the adoption of the new fitness function resulted in superior performance for KCS-LSTM. Consequently, we ascertain that the implementation of the new fitness function is meritorious.

We also compare the prediction results of several models to validate the comprehensive performance of the proposed method, including the traditional models like BP and RNN

**Table 7  The process of change in the value range of hyperparameters.**

| Iteration number | Range of hyperparameter values | | | | |
|---|---|---|---|---|---|
| | Learning rate | Epoch | nH1 | nH2 | Time step |
| 0 | [0.001,0.01] | [10,100] | [1,150] | [1,150] | [2,60] |
| 10 | $[5.90 \times 10^{-3}, 7.12 \times 10^{-3}]$ | [10,35] [60,100] | [1,38] [53,150] | [1,109] | [2,17] [37,60] |
| 20 | $[5.82 \times 10^{-3}, 6.40 \times 10^{-3}]$ $[6.53 \times 10^{-3}, 6.78 \times 10^{-3}]$ $[6.87 \times 10^{-3}, 7.12 \times 10^{-3}]$ | [90,100] | [1,38] [53,83] [116,150] | [23,41] | [7,12] |
| 30 | $[6.13 \times 10^{-3}, 6.40 \times 10^{-3}]$ $[6.53 \times 10^{-3}, 6.78 \times 10^{-3}]$ $[6.87 \times 10^{-3}, 7.12 \times 10^{-3}]$ | [90,92] [95,100] | [1,24] [53,83] [116,137] | [30,33] | [7,9] [10,12] |
| 40 | $[6.57 \times 10^{-3}, 6.75 \times 10^{-3}]$ | [90,91] [95,98] [98,100] | [116,130] [132,137] | [30,32] [32,33] | [10,11] |
| 50 | $[6.65 \times 10^{-3}, 6.75 \times 10^{-3}]$ | [90,91] [95,97] [98,99] | [116,124] [125,130] [134,137] | [30,31] [31,32] [32,33] | [10,11] |

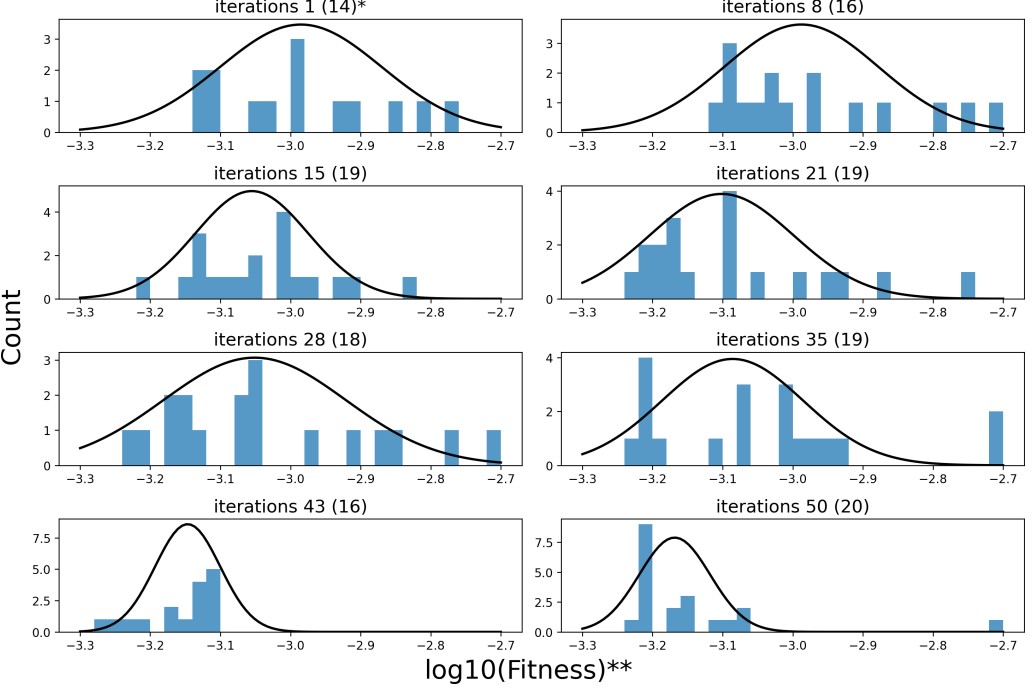

**Figure 7  The process of fitness distribution changes in the search space.** *(A)* * iterations 1(14) denotes the distribution of 14 sample points in the first iteration of search. (B)** We have made an adjustment to fitness to make the results more intuitive: $x = log10(Fitness)$.

**Table 8   Results of hyperparametric optimization.**

| Parameter | Search range | Optimal value |
|---|---|---|
| Learning rate | [0.001,0.01] | 0.006718 |
| Epoch | [10,100] | 95 |
| Neuron numbers in the LSTM layer | [1,150] | 120 |
| Neuron numbers in the fully connected layer | [1,150] | 30 |
| Time step | [2,60] | 11 |

**Table 9   Comparison of prediction measures of different models.**

| Model | Metric | | | | | |
|---|---|---|---|---|---|---|
| | MSE | MAE | sMAPE | RSE | CORR | SA |
| KCS-LSTM | 4.1432 | 1.0966 | 0.0216 | 0.0691 | 0.99762 | 0.4949 |
| ACO-LSTM | 4.3614 | *1.2492* | *0.0245* | 0.0709 | *0.99777* | 0.4039 |
| KCS-RNN | 5.0949 | 1.2717 | 0.0247 | 0.0767 | 0.99724 | 0.4479 |
| ACO-RNN | 4.6749 | 1.2516 | *0.0245* | 0.0734 | 0.99759 | 0.4160 |
| ACO-BP | 6.5208 | 1.6372 | 0.0327 | 0.0867 | 0.99660 | 0.4387 |
| BP | 5.7137 | 1.4748 | 0.0282 | 0.0812 | 0.99714 | *0.4532* |
| Informer | 19.1286 | 3.5976 | 0.0782 | 0.1483 | 0.99602 | 0.3584 |
| LSTNet | 15.0160 | 3.1355 | 0.0625 | 0.1315 | 0.99677 | 0.3717 |
| SCINet | *4.3147* | 1.2676 | 0.0248 | *0.0705* | 0.99752 | 0.4444 |
| KCS+LSTM | 11.5543 | 2.8504 | 0.0622 | 0.1155 | 0.99782 | 0.3568 |
| IMP | 4.0% | 12.2% | 11.9% | 2.0% | – | 9.2% |

**Notes.**
[a]The best results are highlighted with bold underline and second best results are shown in italic bold.
[b]IMP shows the improvement of KCS-LSTM over the best model.

for TSF, a variant of RNN called Long- and Short-term Time-series network (LSTNet) (*Lai et al., 2018*), a variant of Transformer called Informer, and a variant of TCN called SCINet. They cover the primary classes of artificial neural networks in the field of TSF and will be used as baseline models to validate the predictive capability of our models. The main experimental results are shown in Table 9 and Fig. 9. In the task of short-term time series forecasting of arbitrage spreads, KCS-LSTM demonstrates superior performance compared to other time series forecasting models.

To be specific, BP is an early artificial neural network model applied to time series forecasting. In the experiment, the same hyperparameters were used as those optimized for the LSTM model by the KCS algorithm. However, from the prediction curves and evaluation metrics, the prediction of BP is not accurate enough. It validates that the structure of LSTM has a superior advantage in simulating spread data. As for the RNN model, which is the earliest form of recurrent neural network, we also conducted hyperparameter optimization on it using the KCS algorithm. KCS-LSTM still has better performance. Moreover, we have also tested LSTNet, Informer, and SCINet on the arbitrage spread dataset. As can be seen from Table 9, compared with a transformer-based method like Informer, the LSTNet method based on RNN and 2D convolution produces better forecasting results.

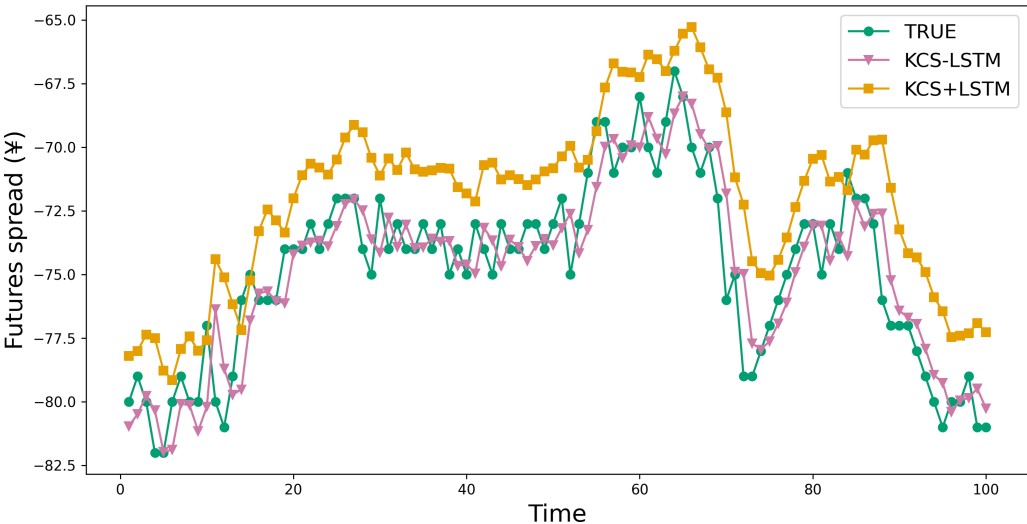

**Figure 8** **Comparison between prediction results of KCS-LSTM and KCS+LSTM.** KCS+LSTM network utilizes the MSE value of the validation set as its fitness function and integrates two dropout layers within its LSTM network.

One of the primary reasons (*Greff et al., 2016*) is that, for short-term forecasting, the recent data points are typically more crucial for accurate forecasting. However, the permutation invariant self-attention mechanisms used in Transformer-based methods do not pay much attention to such key information. In contrast, the general sequential model (RNN/TCN) can easily formulate it. As another prediction method, SCINet outperforms the LSTNet method. Finally, it is worth noting that KCS-LSTM outperforms BP, KCS-RNN, LSTNet, Informer, and SCINet on MSE by 27.5%, 18.7%, 72.4%, 78.3%, and 4.0%, respectively. Our prediction model achieves the best prediction results. Our model also achieves the best results on MAE, sMAPE, and RSE, with improvements of over 45% compared to LSTNet and Informer, but only 13.5%, 12.9%, and 2.0% on SCINet. It is clear that our model is only slightly better than SCINet. However, the complexity of LSTM is much lower than that of SCINet, so our optimization method is effective. The experimental results of KCS-RNN, which are close to various metrics of SCINet, further validate this conclusion. Furthermore, our conclusions can be supported by the SA, which shows the accuracy of the forecasts in predicting the direction of change compared to the actual spread data. Our model improves by 9.2%, 10.5%, 33.1%, 38.1%, and 11.4% compared to other prediction models. This indicates that our prediction model is superior in determining the direction of spread change. Finally, we also use the traditional ACO algorithm to optimize the LSTM network. Our model improves the MSE by 5%. When optimizing the RNN, ACO improves by 8.2% compared to KCS. Both models are relatively small, but ours is more advantageous in selecting the number of populations and iterations.

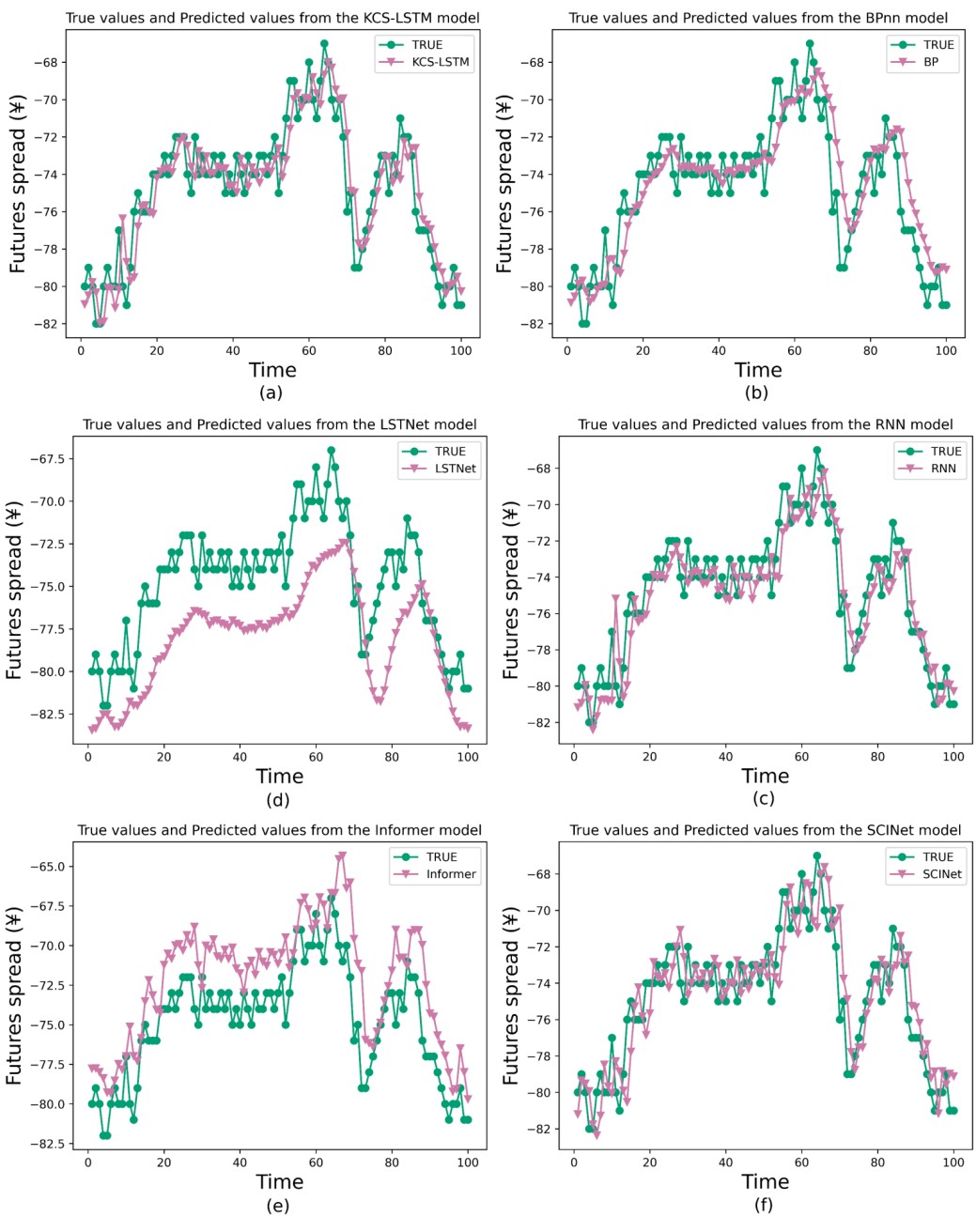

**Figure 9** Comparison between prediction results of different model.

## CONCLUSION

This article presents a novel network for predicting arbitrage spread movements called K-fold Cross-Search optimized Long Short-Term Memory (KCS-LSTM). The network uses the K-fold Cross-Search (KCS) algorithm to adaptively tune the hyperparameters of the LSTM network. This process reduces the influence of personal factors on the

predictive model and enhances its accuracy. The random search in the KCS hyperparameter optimization algorithm is based on the pheromone feedback mechanism of the ACO algorithm. The method introduces a transformation mechanism for the search space and an iterative updating mechanism, reducing reliance on population size and the randomness in the number of iterations required. The novel algorithm will approach the global optimal solution by gradually eliminating unpromising regions. In addition, the novel fitting functions are designed to enhance the generalisation ability of the hyperparametric optimization network. Additionally, this article evaluates the performance of KCS-LSTM using actual arbitrage spread data for rebar and hot-rolled coil. In the analysis of six quantitative metrics (*i.e.,* MSE, MAE, sMAPE, RSE, CORR, and Symbolic Accuracy), the comparison with several mainstream artificial neural network models in time series forecasting shows that the proposed KCS-LSTM network has high effectiveness in predicting the movement of arbitrage spreads.

In an increasingly competitive arbitrage market, the KCS-LSTM model can effectively exploit the forecasting potential of LSTM and achieve high-precision arbitrage spread prediction, which can improve the competitiveness of arbitrage strategies. The ability of network to efficiently handle time series data has a certain application value for other time series problems. Furthermore, there are some limitations that need to be considered in future research. Firstly, the model proposed in this article can achieve short-term forecasting of arbitrage spread, but its medium-term and long-term forecasting abilities need improvement. Secondly, this article considers statistical indicators of predicted outcomes but does not study financial indicators when combined with trading strategies. Therefore, in the upcoming research, additional network structures can be explored to enhance the long-term prediction of arbitrage spread. Future predictive models will be validated through live trading with the trading strategy.

### Funding

This work was supported by the Henan University Science and Technology Innovation Team Support Plan (20IRTSTHN013) and Henan Province Key R&D and Promotion Special Project (212102210166). The funders had no role in study design, data collection and analysis, decision to publish, or preparation of the manuscript.

### Grant Disclosures

The following grant information was disclosed by the authors:
Henan University Science and Technology Innovation Team Support Plan: 20IRTSTHN013.
Henan Province Key R&D and Promotion Special Project: 212102210166.

### Competing Interests

Panke Qin is employed by Hebi National Optoelectronic Technology Co, Ltd.

## Author Contributions

- Zeliang Zeng conceived and designed the experiments, performed the experiments, analyzed the data, performed the computation work, prepared figures and/or tables, authored or reviewed drafts of the article, and approved the final draft.
- Panke Qin conceived and designed the experiments, analyzed the data, authored or reviewed drafts of the article, and approved the final draft.
- Yue Zhang conceived and designed the experiments, authored or reviewed drafts of the article, and approved the final draft.
- Yongli Tang conceived and designed the experiments, authored or reviewed drafts of the article, and approved the final draft.
- Shenjie Cheng performed the experiments, authored or reviewed drafts of the article, and approved the final draft.
- Sensen Tu performed the experiments, authored or reviewed drafts of the article, and approved the final draft.
- Yongjie Ding performed the experiments, authored or reviewed drafts of the article, and approved the final draft.
- Zhenlun Gao performed the experiments, authored or reviewed drafts of the article, and approved the final draft.
- Yaxing Liu performed the experiments, authored or reviewed drafts of the article, and approved the final draft.

## Data Availability

The raw data and code are available in the Supplemental Files.

## Supplemental Information

Supplemental information for this article can be found online at http://dx.doi.org/10.7717/peerj-cs.2215#supplemental-information.

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
