# Peer review of "An optimized LSTM network for improving arbitrage spread forecasting using ant colony cross-searching in the K-fold hyperparameter space"

_PeerJ Computer Science, doi:10.7717/peerj-cs.2215_

## Round 0.1 · original submission · Major Revisions

Dear authors,

Thank you for submitting your article. The reviewers’ comments are now available. Your article has not been recommended for publication in its current form. However, we encourage you to address the reviewers' concerns and criticisms; particularly regarding novelty, literature review, readability, quality, experimental design, and validity, and resubmit your article once you have updated it accordingly.

Best wishes,

**Language Note:** PeerJ staff have identified that the English language needs to be improved. When you prepare your next revision, please either (i) have a colleague who is proficient in English and familiar with the subject matter review your manuscript, or (ii) contact a professional editing service to review your manuscript. PeerJ can provide language editing services - you can contact us at [email protected] for pricing (be sure to provide your manuscript number and title). – PeerJ Staff

Reviewer 1 ·

Basic reporting

1-First of all, the authors work on hyperparameter optimization, which is a very useful subject for the scientific world. This is a very sensitive issue.
2-More content about related works should be provided.
3-The motivation for the study is not clear. What is the main reason that pushed you to do this work?

Experimental design

4-Why didn't the authors use the ant colony algorithm?
5-The authors stated that they proposed a new metaheuristic optimization algorithm inspired by the principles of the ant colony algorithm. I request the authors to compare whether their proposed algorithm (KFC) is more successful than the ant colony algorithm using CEC19 benchmarkt test functions.

Validity of the findings

6-The conclusion section should be expanded.

Reviewer 2 ·

Basic reporting

The paper effectively articulates the problem of large-scale model development in arbitrage spread prediction, providing a clear context for the proposed solution.

The paper presents a comprehensive methodology, covering data preprocessing, cointegration analysis, the KCS algorithm, and the KCS-LSTM network. The step-by-step presentation enhances clarity.

Experimental design

The use of real arbitrage spread data for experiments enhances the practical relevance of the proposed model. It demonstrates the applicability of the KCS-LSTM network in a real-world financial context.

The paper provides detailed explanations of key components, such as the KCS algorithm, LSTM network architecture, and hyperparameter optimization. This aids in understanding the intricacies of the proposed approach.

While the paper mentions the use of real arbitrage spread data, it could benefit from a more detailed description of the dataset, including its source, characteristics, and any preprocessing steps performed.

Validity of the findings

Providing more specific details on the results, such as performance metrics and comparative analysis with baseline models, would strengthen the paper's empirical validation.

Ensure consistent use of terminology and avoid undefined abbreviations. For instance, clarify the meaning of "KCS" upon its first mention.

The use of real arbitrage spread data adds credibility to the research. However, the paper lacks a detailed description of the dataset, including its source, characteristics, and preprocessing steps. More details on data properties would strengthen the paper.

The experimental design, involving the KCS algorithm for hyperparameter tuning, is well-presented. However, the paper lacks specific details on performance metrics and comparative analysis with baseline models. Including these details would provide a more concrete evaluation of the proposed model.

Additional comments

The paper introduces an interesting approach but needs improvements in terminology consistency, dataset description, and quantitative analysis to elevate its technical impact. Addressing these aspects would enhance the technical robustness of the work.

Reviewer 3 ·

Basic reporting

The application introduced in the paper is intriguing; however, the novelty of the LSTM RNN hyperparameter optimization method is not clear.

The article falls short in referencing related work on applying ACO (Ant Colony Optimization) to ANN (Artificial Neural Network) hyperparameter tuning, which is readily available.

The paper contains typographical and editorial errors, such as 'Error! Reference source not found' and 'Search Spare' (the reader can assume that the authors meant 'search space'?).

The quality of Figure 3 is insufficient for readers to easily comprehend.

Sections 3.1.1 and 3.1.2 are not clear and hard to follow. Also, the reader can find it useful if the explanation in the text went along with the flowchart (maybe add the step number beside/inside the related flowchart block).

The flowchart provided does not adhere to the standards of formal notation, potentially making it challenging to understand. For example, the yes-no arrows from the predicate blocks intersect. Additionally, it's unclear if the training/testing process of the LSTM models differs from the 'calculate loss function' process, which includes backpropagation according to the flowchart.

The authors use the phrase 'the precision of the hyperparameter,' but its precise meaning remains unclear.

Experimental design

no comment

Validity of the findings

no comment

Additional comments

no comment

---

## Round 0.2 · Major Revisions

Dear authors,

Thank you for your submission. Your article has not been recommended for publication in its current form. However, we do encourage you to address the concerns and criticisms of the reviewers and resubmit your article once you have updated it accordingly.

Best wishes,

· Appeal

Appeal

Thanks for your letter in response to our submission of our manuscript (An optimized LSTM network for improving arbitrage spread forecasting using ant colony cross-searching in the K-fold hyperparameter space). After carefully studying your decision letter and the 2nd-round comment from reviewer 3, we realize that the novelty of our work were not fully identified or recognized by the reviewers. Here we would emphasize that the most notable merits of our manuscript include:

1)We propose a novel KCS method for hyperparameter optimization.

2)We have used the KCS-LSTM network in a more recent scenario, the area of arbitrage spread prediction.

Specifically, reviewer 3 had mentioned the following question in the 2nd round of comments:

The reviewer finds it challenging to identify a significant novelty in the work presented, particularly from a neural optimization perspective. The concept of employing Ant Colony Optimization (ACO) for neural hyperparameters has been explored previously, and the following are just a few examples:

Although this paper does not make significant changes to the structure and process of the ANN optimization model, it presents a redesigned optimization algorithm. The novel algorithm differs significantly from the previously studied ACO in terms of search ideas.

In the field of hyperparameter optimization, both the current ACO algorithms and their improved versions are designed to search the entire hyperparameter space by exploring new regions continuously through certain mechanisms. Determining the necessary number of iterations can be challenging, leading to the issue of repeated searches.

However, our KCS algorithm proposes a new angle for the optimization approach by gradually optimizing the entire search space through the continuous elimination of certain regions using a specific mechanism. This approach combines traditional grid search with ACO’s pheromone mechanism. The traversal process is transformed from a sequential structure to a tree structure by changing the search unit from a coordinate point to a block of intervals. Each node in the tree represents a finite search space. The search process involves selecting an iterative update path of the search space from the root node to the leaf nodes. The selection process relies on two update mechanisms in the search space and the pheromone mechanism of the ACO. The depth of the entire tree is primarily affected by the size of each elimination interval in the worst elimination rule. This approach restricts the complexity of the problem that the pheromone search mechanism needs to tackle in single-round search. And determining the number of iterations required can be done more easily, thus avoiding ineffective searches.

Regarding the optimization algorithm, the reviewer 1 has made a relevant comment and did not raise any further questions in the 2nd-round comment.

We understand that the misunderstanding might be caused by the unclear description in our manuscript and response letter, but we believe that the method is of merit and the paper is potentially publishable in the journal.

We look forward to hearing from you again.

Thank you again for your time and effort in processing our manuscript.

Best regards,

Zeliang Zeng


· · Academic Editor

Reject

Dear authors,

One reviewer has significant concerns regarding the quality, novelty, and originality of your paper. The paper also still appears to have insufficient fundamental theoretical information and experimental details. It seems that you do not provide sufficient evidence for the novelty of the presented method. There is also not sufficient reference to related work that addresses the cornerstone of the novelty of the work on ANN hyperparameter optimisation using ACO. It is also difficult to justify the novelty of the work solely from the point of view of optimization, In view of the criticisms of expert reviewers, unfortunately, your manuscript has not been recommended for publication.

Best wishes,

Reviewer 1 ·

Basic reporting

It seems that the authors reflected all the requirements in the article, including the CEC19 tests requested from them.

Experimental design

It seems that the authors reflected all the requirements in the article, including the CEC19 tests requested from them.

Validity of the findings

It seems that the authors reflected all the requirements in the article, including the CEC19 tests requested from them.

Additional comments

It seems that the authors reflected all the requirements in the article, including the CEC19 tests requested from them.

Reviewer 3 ·

Basic reporting

The reviewer finds it challenging to identify a significant novelty in the work presented, particularly from a neural optimization perspective. The concept of employing Ant Colony Optimization (ACO) for neural hyperparameters has been explored previously, and the following are just a few examples:


Bacanin, N., Bezdan, T., Tuba, E., Strumberger, I. and Tuba, M., 2020. Optimizing convolutional neural network hyperparameters by enhanced swarm intelligence metaheuristics. Algorithms, 13(3), p.67.

Trajkovski, A. and Madjarov, G., 2022. Model Hyper Parameter Tuning using Ant Colony Optimization.

Jlassi, S., Jdey, I. and Ltifi, H., 2021. Bayesian Hyperparameter Optimization of Deep Neural Network Algorithms Based on Ant Colony Optimization. In Document Analysis and Recognition–ICDAR 2021: 16th International Conference, Lausanne, Switzerland, September 5–10, 2021, Proceedings, Part III 16 (pp. 585-594). Springer International Publishing.

Experimental design

no comment

Validity of the findings

no comment

Reviewer 4 ·

Basic reporting

Basic Reporting
The authors proposed a K-fold Cross-Search algorithm-optimized LSTM network for arbitrage spread prediction. The KCS-LSTM network was validated using real spread data of rebar and hot-rolled coil from the past three years. The KCS-LSTM network is shown to be competitive in predicting arbitrage spreads compared to complex neural network models. However, there are few weaknesses which are highlighted below:
The structure of the article is coherent. However, there are few short comings. In the Introduction Section, I did not find proper motivation to conduct the research. Secondly, mostly old references have been used. On Line 82, four kinds of methods for time series modeling have been discussed but no comparison / justification has been given for selected model. It is not clear that why LSTM has been selected? Why not Bi-LSTM, GRU and bi-GRU have been used? From Line 145 to 160, the text should be concise which should increase the readers’ readability. On Line 161, it is mentioned that “The remaining Chapters”, do rectify. Section 2.1 addresses the problem statement. In my view, the problem should be highlighted with the support of review of literature. Not a single reference has been cited in the problem statement. Therefore, I have already commented that the references used are quite old. Problem Statement and Data Analysis should not be heading. It should be merge with introduction. Section 2.2 related to data structure should be moved to the methodology section and methodology diagram should be there to reflect the flow of research. A picture is more worth than words so, use more illustrative figures and diagrams to visually represent the architecture and workflow of the proposed models. Overall, the paper needs re-arrangement.

Experimental design

The experimental design fit the aims and scope of the paper. The author can also design evaluation methodology to increase the readability of the results.

Validity of the findings

Lastly, the conclusion section lacks a detailed description of future research and shortcomings of the paper. Provide insights into potential future research directions and areas for improvement based on the findings of the current study.

Additional comments

Language and Style:
Ensure consistency in terminology and writing style throughout the article.
Proofread the manuscript for grammatical errors, typos, and awkward phrasings to improve readability.

Annotated reviews are not available for download in order to protect the identity of reviewers who chose to remain anonymous.

---

## Round 0.3 · accepted · Accept

Dear authors,

Thank you for the revision and for clearly addressing all the reviewers' comments. I confirm that the paper is improved. Your paper is now acceptable for publication in light of this revision.

Best wishes,

Reviewer 4 ·

Basic reporting

No comments

Experimental design

No comments

Validity of the findings

No comments